# On a Built-in Conflict between Deep Learning and Systematic Generalization

## Abstract

Out-of-distribution or systematic generalization is a desirable property that most deep learning algorithms lack. In this paper, we hypothesize that internal function sharing is one of the reasons to weaken systematic generalization in deep learning for classification tasks. Under equivalent prediction, a model partitions an input space into multiple parts separated by boundaries. The function sharing prefers to reuse boundaries, leading to fewer parts for new outputs, which conflicts with systematic generalization. We show such phenomena in standard deep learning models, such as fully connected, convolutional, residual networks, LSTMs, and (Vision) Transformers. We hope this study provides novel insights and forms a basis for new research directions to improve systematic generalization. Source codes are available in the supplementary material.

## 1 Introduction

A fundamental property of artificial intelligence is generalization, where a trained model appropriately processes unseen test samples. Many problems adopt the i.i.d. assumption. On the other hand, out-of-distribution (o.o.d.) or *systematic generalization* (Fodor & Pylyshyn, 1988; Lake & Baroni, 2018) requires that training and test distributions are disjoint so that test samples have zero probability in training distribution. Systematic generalization is crucial for human learning and supports efficient data use and creativity. Therefore, machines are encouraged to acquire such generalization ability to achieve human-like intelligence.

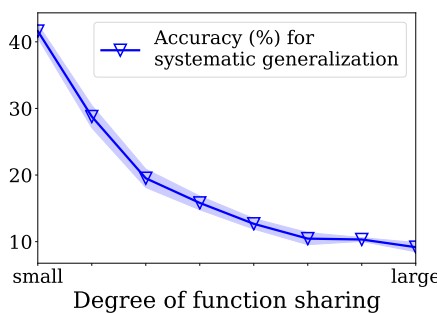

Figure 1: A simplified plot for a result in the experiment section. As the degree of function sharing increases (more sharing layers), the accuracy of the test dataset or generalization capacity decreases accordingly. It indicates that function sharing in deep learning conflicts with systematic generalization. Please refer to Section 3 and Figure 5a for details.

Systematic generalization usually requires that a sample has multiple explanatory factors of variation (Bengio et al., 2013), and the generalization is enabled by producing an unseen combination of seen factor values. For example, models trained on blue rectangles and green triangles predict blue triangles. We adopt factors mainly in designing experiments and developing intuitions. It helps experiments because new outputs are only related to function sharing between factors (Section 3). So we limit our claim to the cases for recombination of factors.

One stream of artificial intelligence is Connectionism (Feldman & Ballard, 1982; Rumelhart et al., 1986), which uses many simple neuron-like units richly interconnected and processed in parallel. It was criticized that Connectionist models do not support systematic generalization well (Fodor & Pylyshyn, 1988; Marcus, 1998). *Deep learning* (LeCun et al., 2015) originates from Connectionism, and various techniques have enabled multiple-layer modelings and improved performance on i.i.d. problems in recent years. Also, specific algorithms have been proposed to equip deep learning with systematic generalization ability (Russin et al., 2019; Lake, 2019). However, less discussion has been made on why standard deep learning models do not achieve systematic generalization.

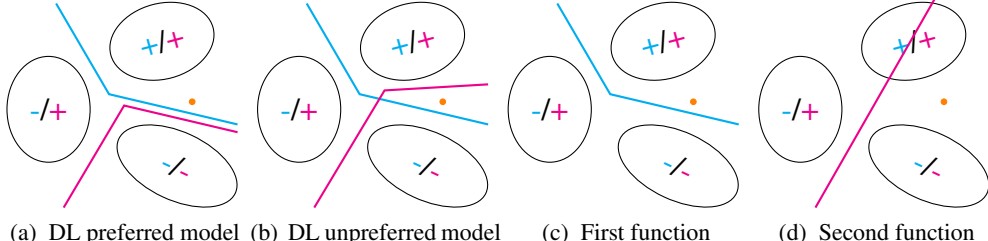

(a) DL preferred model  (b) DL unpreferred model  (c) First function  (d) Second function

Figure 2: Intuitions for the model preference of deep learning (DL). Each figure is an input space containing three sets of training samples with outputs of +/+, -/-, and -/+, respectively. Models have two decision boundaries (cyan and magenta). The orange dot is a test sample with a new ground-truth factor combination +/-. We discuss that the training process prefers the first case (a) over the second one (b) because it likes to share or reuse functions. Suppose the first function (c) is learned, then the process tends to reuse it by learning a simple function (d) and combining them instead of learning the complicated magenta function in (b) from scratch. So the function between +/+ and -/- is shared. Few inputs are mapped to the new output, and systematic generalization is not achieved.

This paper addresses the above question by looking into a built-in characteristic of deep learning. A node (or a set of nodes) is a function of the network input. By *function sharing*, we mean nodes in one layer share inputs from nodes in the previous layer. It may also be called activation sharing or feature sharing. We hypothesize that function sharing is one of the reasons to prevent systematic generalization in deep learning. Under equivalent prediction, a classification network partitions an input space into multiple parts separated by boundaries. Function sharing prefers to reuse boundaries and avoid redundant ones for training predictions. It leads to fewer parts for new outputs and weakens systematic generalization. The nearest neighbor classifier is an analogy for this effect because it predicts a training sample output for any test sample, so it does not have a part for new outputs. We also discuss that function sharing belongs naturally to deep learning (Section 4).

Figure 2 has an intuitive example explaining why the conflict happens. A test sample (in orange) equals a set of training samples (+/+) on

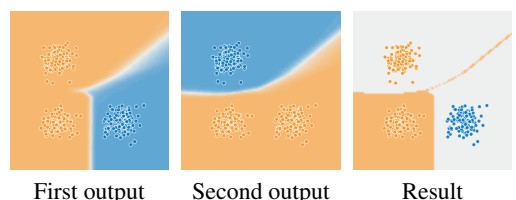

First output     Second output     Result

Figure 3: A visualized example with a two-dimensional input and two binary outputs. The model is a multiple-layer fully connected neural network. The three figures contain the predictions for the first and the second outputs and classification results for the model. Each figure is the input space. The dots are training samples whose colors are ground-truth labels. For output figures, the background colors are model predictions. We set the new combination of labels (blue-blue) as the correct pair of outputs. For the result figure, the background is orange if both predictions are wrong, white if one is correct, and blue if both are correct. The result figure does not have a blue background, indicating that the model does not predict the new label combination for any input.

the first output. Then they are also equal on the second output if the function is reused (see caption). Therefore, they are equal in all the outputs, which conflicts with systematic generalization. More generally, the two boundaries jointly partition an input space into multiple parts, and deep learning prefers (a) because it has fewer parts than (b). Figure 3 has a visualized example. It is similar to the example in Figure 2a. The two functions are shared in the top-right region, and few inputs are predicted as the new combination. Figure 1 has a simplified plot for a result in the experiment section. As the degree of function sharing increases (more sharing layers), the accuracy of the test dataset or generalization capacity decreases accordingly. It supports that function sharing weakens systematic generalization. Please refer to Section 3 for more details.

This paper contributes to uncovering a built-in conflict between deep learning and systematic generalization. We hope this study provides novel insights, forms a basis for new research directions, and helps improve machine intelligence to the human level.

## 2   A BUILT-IN CONFLICT

We hypothesize a built-in conflict between the function sharing in deep learning and systematic generalization. We cover the definition, assumptions, and derivations of the conflict.

### 2.1   SYSTEMATIC GENERALIZATION

We have input set $X$ and output set $Y$. $Y$ is decided by $K$ factors $Y_1, \ldots, Y_K$. The training set $\mathcal{D}_{\text{train}}$ has input $X_{\text{train}}$ and output $Y_{\text{train}}$. The test set $\mathcal{D}_{\text{test}}$ has input $X_{\text{test}}$ and output $Y_{\text{test}}$. In systematic generalization, $Y_{\text{train}}$ and $Y_{\text{test}}$ are disjoint. However, the values for each factor $i$ are included in the training output. A model $f$ maps input $X$ to the prediction of output $f(X)$. A model enables systematic generalization if it correctly predicts the ground-truth test outputs.

**Definition 1** (Systematic generalization). *The dataset requires that any test label is not a training label, but each factor of a test label is seen in a training label.*

$$\forall (x, y) \in \mathcal{D}_{test} : y \notin Y_{train}, \quad \forall i = 1, \ldots, K, \exists y' \in Y_{train} : y_i' = y_i$$

*A function $f$ enables systematic generalization if $\forall (x, y) \in \mathcal{D}_{test} : y = f(x)$.*

When the model is well trained, we assume it correctly predicts training samples.

**Assumption 1** (Correct training prediction). $\forall (x, y) \in \mathcal{D}_{train} : y = f(x)$.

### 2.2   FUNCTION SHARING

In Figure 2, deep learning prefers the model in (a) over the model in (b). We denote the models as $f$ and $g$ (mapping from input to output), respectively. We see that $g$ has one more region for the new factor combination between the magenta and cyan boundaries. This region exists because the magenta boundary in $g$ splits the +/+ region in $f$ into two parts. It means the partition under $g$ refines that under $f$, but not vice versa.

We assume a property of function sharing. For general models $f$ and $g$, deep learning prefers $f$ over $g$ more or equally if the partition under $g$ is a refinement of that under $f$. Equivalently, if two inputs have an identical prediction in $g$, their predictions are still equal in $f$. It means an equal prediction in $g$ implies that in $f$.

**Assumption 2** (Function sharing). *Deep learning prefers $f$ over $g$ more or equally if*

$$\forall x_a, x_b \in X : g(x_a) = g(x_b) \implies f(x_a) = f(x_b)$$

Note that it is a bias in the learning process. In case we need $f$ to be strictly more preferred over $g$, we can additionally show that identical prediction in $f$ does not imply that in $g$.

We consider what causes the function sharing. Intuitively, the preference comes from greedy optimization and the task requirement to learn a complicated model. A model learns to split or partition inputs by different predictions. Some splits (can be a factor) might be easier to learn than others. For example, learning to split inputs by color is more straightforward than by shape. Then the greedy optimization learns it quickly and reuses its intermediate functions to learn other splits. It is not likely that multiple splits (or factors) are learned equally fast during the whole training process.

The mechanism of reusing function is similar to auxiliary tasks. The parameters are updated to address a complicated main task while quickly learning and keeping the prediction ability for more straightforward auxiliary tasks. A more common but less similar example is pre-training. Pre-trained modules, such as image feature extractors or word embeddings, share the internal functions with the main target tasks. However, the pre-training and the main task training do not happen simultaneously. We will discuss more insights in Section 4 and Appendix C.

### 2.3   THE CONFLICT

We derive propositions for proving the theorem and explaining the reasons for the phenomena. In practice, the assumptions may not hold strictly, and the effects come more from the soft biases of the conclusions. The proofs are in Appendix B.

Assumption 2 leads a model to predict training outputs for any input.

**Proposition 1** (Seen prediction). *From Assumption 2, $\forall x \in X : f(x) \in f(X_{train})$.*

Informally, suppose there is a non-empty set of all inputs that a function $f$ does not map to training outputs $f(X_{\text{train}})$. In that case, we can design another function $f'$ that predicts a training output for these inputs and keeps predictions for other inputs. Then both $f$ and $f'$ equivalently distinguish training outputs. With Assumption 2, $f'$ is preferred, hence $\forall x \in X : f(x) \in f(X_{\text{train}})$. It does not apply Assumption 1, indicating that the phenomena may happen before a model is well trained.

If a model performs well for the training set, it predicts training ground-truth output for any input.

**Proposition 2** (Seen label). *From Assumption 1 and Proposition 1, $\forall x \in X : f(x) \in Y_{train}$.*

It says that the prediction is not any new output. It is a stronger argument than avoiding a particular new output for each input in systematic generalization. It explains that prediction is incorrect because any new output is resisted. We evaluate it in Section 3.

We then look at the conflict. For any test sample $(x, y)$, the definition of systematic generalization requires that the output $y$ is not a training output $Y_{\text{train}}$. However, this contradicts Proposition 2.

**Theorem 1** (The conflict). *From Definition 1 and Proposition 2, $\forall (x, y) \in \mathcal{D}_{test} : y \neq f(x)$.*

We covered systematic generalization definition and function sharing assumption. We then derived propositions and a theorem of the conflict.

## 3 EXPERIMENTS

We run experiments to show that function sharing reduces the ability of systematic generalization in deep learning. We not only compare sharing or not but also adjust the degree of sharing. We focus on the cases where new outputs are unseen combinations of seen factor values. We cover different standard deep neural network models. The details of networks and experiments can be found in Appendix D. More experiments with natural inputs are in Appendix A. The results of zero-shot learning datasets are in Appendix E, where factors are mainly related to the input locality. We look at the experiment settings and results.

### 3.1 SETTINGS

**Data preparation**   We construct a dataset from two ten-class classification datasets. The training data are generated from the original training dataset. We first chose output $y$ and chose $x$ based on it. $y_1$ is chosen from all possible labels. $y_2$ is chosen from five classes $\{y_1, y_1 + 1, \ldots, y_1 + 4\}$ (we use modular for labels). The test data are generated from the original test dataset. $y_1$ is chosen in the same way as in training. $y_2$ is chosen from the other classes $\{y_1 + 5, y_1 + 6, \ldots, y_1 + 9\}$. In this design, training and test label combinations are mutually exclusive, but test labels for each output factor are seen in training. Any factor label appears evenly in training and test combinations. $x_1$ and $x_2$ are chosen conditioned on their labels $y_1, y_2$, respectively, and merged as the input $x$. The original datasets and input merge methods vary for each experiment. All the choices follow uniform distributions.

**Architecture**   To evaluate the influence of function sharing on new outputs, we change the function sharing ability while keeping other properties stable for a deep learning model. So we modify function sharing related to new outputs. Since a new output appears only as a new factor combination in our setting, we adjust the sharings between output factors. It does not need to remove all sharings and avoid difficulties in model design. We choose a layer and duplicate the following layers, keeping the number of all hidden nodes in each layer if feasible (Figure 4). We call the former part a shared network and the latter parts individual networks. Each individual network predicts one output factor, so the output is disentangled. We will discuss entangled outputs in Section 4. We keep the depth of the whole network and change the depth of the shared network and individual networks. Note that only sharing the input layer indicates learning two separate models.

**Evaluation metrics**   We use accuracy as the metric. A sample prediction is correct if all the outputs are correct. We have three types of accuracy. The first is the regular evaluation of test data

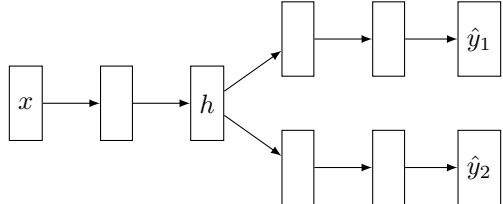

Figure 4: Deep model architecture. $x$ is input, and $f(x) = \hat{y}_1, \hat{y}_2$ are outputs. We duplicate the layers after layer $h$. The shared network is up to $h$, and the individual networks are after $h$ (their depths are all two here). We change the shared network depth while maintaining the entire network depth so that the function sharing ability is adjusted while other network properties are stable.

for systematic generalization (a: **Test Sample Accuracy**) corresponding to Theorem 1. We also consider a set of inputs mapped to unseen output combinations corresponding to Proposition 2. We evaluate whether the test samples predict one of the unseen output factor combinations (b: **Test Set Accuracy**). However, test samples are only a subset of input space. If a model learns a different set of factors, the expected inputs may not be those of test samples. So we also evaluate systematic generalization as a model property for any valid input. We randomly draw test inputs from the whole input space (c: **Random Set Accuracy**)[1]. We run each experiment five times and plot the mean and the standard deviation (Figure 5). The result numbers are in Table 2 (Appendix D.3).

$$a: \ \mathbb{E}_{(x,y)\sim P(\mathcal{D}_{\text{test}})}[\delta(f(x) = y)] \quad b: \ \mathbb{E}_{x\sim P(X_{\text{test}})}[\delta(f(x) \in Y_{\text{test}})] \quad c: \ \mathbb{E}_{x\sim U[\mathcal{X}]}[\delta(f(x) \in Y_{\text{test}})]$$

## 3.2 RESULTS

**Fully Connected Network** We use an eight-layer fully connected neural network with a flattened image input. We use the Fashion dataset (Xiao et al., 2017) and the MNIST dataset (LeCun et al., 1998). The datasets are uncomplicated to avoid the training data under-fitting for a fully connected neural network. We merge the two inputs by averaging values at each input node.

**Convolutional Network** We use a convolutional neural network with six convolutional layers and two fully connected layers. We use the CIFAR-10 dataset (Krizhevsky, 2009) and the Fashion dataset (Xiao et al., 2017). We scale the input sizes to that of the larger one and broadcast gray images to colored ones. We merge the inputs by averaging at each node. We use the Fashion dataset as one factor because the average of two colored images can cause training data under-fitting for convolutional neural networks.

**Residual Network** We use ResNet50 (He et al., 2016), which has five stages, each treated as a layer while changing the shared network depth. It has the dataset setting in the CNN experiment.

**Vision Transformer** We use Vision Transformer (Dosovitskiy et al., 2021) with one fully connected layer for each patch, five attention layers, and two fully connected layers. We treat the patches as one layer. It has the dataset setting in the CNN experiment.

**LSTM** A recurrent network has the same parameters for each layer, so it does not support learning different individual networks. Instead, we treat an LSTM as a layer. We use stacked LSTM models with an embedding layer, five bidirectional LSTM layers, and two fully connected layers. We use the Reuters dataset (Joachims, 1998) for both the first and the second datasets. We filter samples by a maximum input length of 200 and use the most frequent ten classes of samples. We merge inputs by concatenating two input texts. The inputs have different lengths because the text lengths vary.

We also run experiments for one-layer LSTM, which compares sharing or not sharing all layers. The results indicate that the shared network has less generalization than the individual network (Table 2).

---

[1]$\delta(\cdot)$ is 1 if the statement is true and 0 otherwise. $U[\mathcal{X}]$ is the uniform distribution of valid inputs.

**Transformer**   We use Transformer (Vaswani et al., 2017). Since it is a classification problem, we only use the encoder. It has one embedding layer, five hidden layers, and two fully connected layers. We use the same dataset setting as the LSTM experiment.

**Summary of results**   Figure 5 shows that, for each evaluation, the accuracy on the left end (not sharing any hidden layer) is higher than that on the right end (sharing all hidden layers), and it generally decreases as the shared network depth increases. The results indicate that the function sharing weakens systematic generalization.

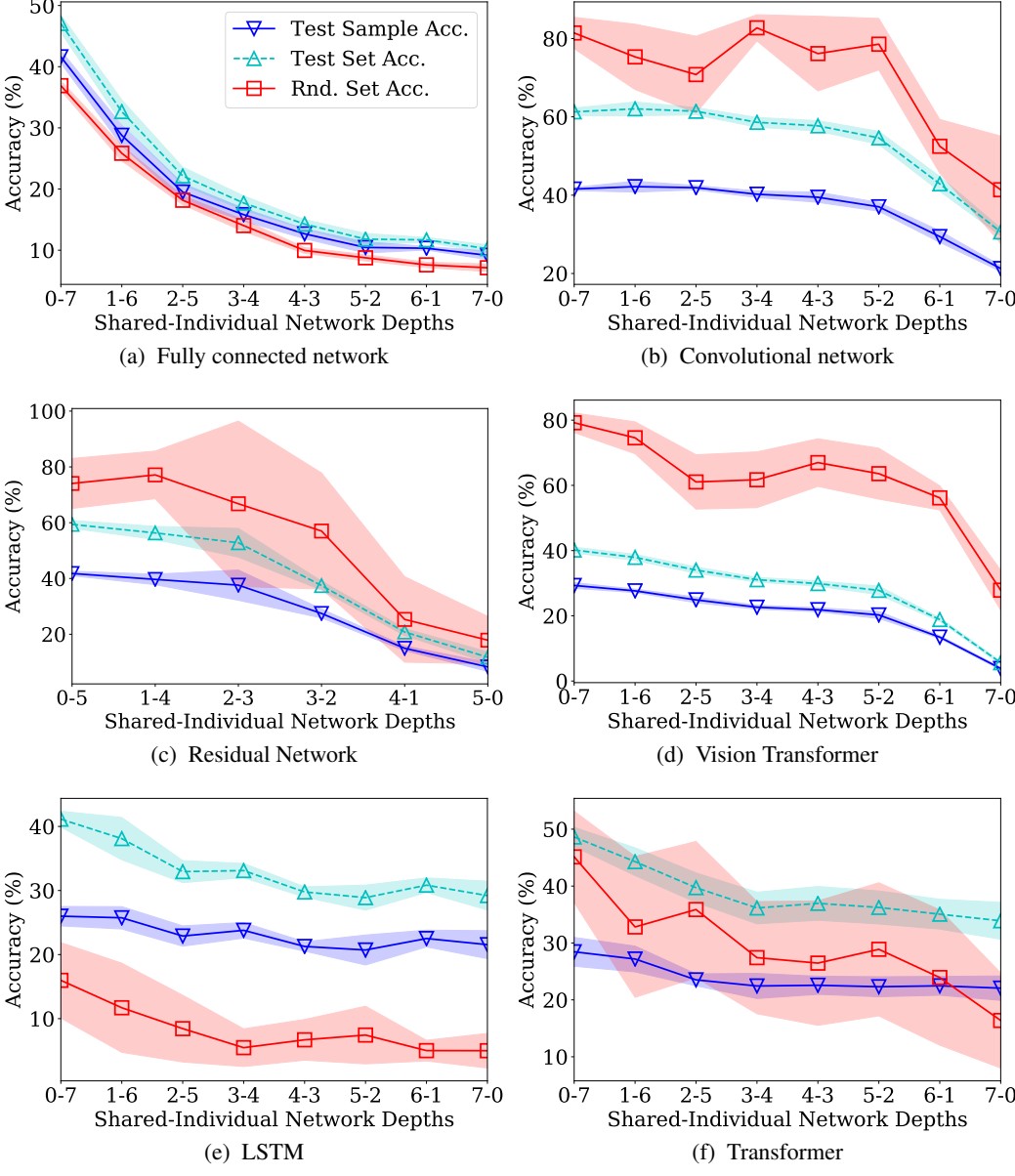

Figure 5: Results of three types of systematic generalization accuracy with shared and individual (sub)network depths. As the shared network depth increases, the accuracy generally decreases, indicating that function sharing reduces systematic generalization ability. Test Set Accuracy and Random Set Accuracy are the ratios of unseen factor combinations in prediction. Test Sample Accuracy and Test Set Accuracy are computed with the test data, and Random Set Accuracy is with randomly generated inputs.

## 4 DISCUSSIONS

### 4.1 ENTANGLED OUTPUT

Though the theory does not require disentangled output, the experiments use it to split the network into individual ones. We discuss that entangled output prediction is not easier than disentangled one. Hence the experiment conclusions will likely extend to entangled outputs. (1) Disentangled output is a particular and usually less complicated case of entangled output. (2) Entangled output can be seen as another shared layer, and the experiments show that increasing shared layers reduces systematic generalization ability, so entangled outputs are also likely to suffer from the problem. (3) We can see each output node of an entangled output as a factor, so it becomes disentangled output. The generalization is even more difficult if a value is unseen for an output node.

### 4.2 BEYOND FACTOR RECOMBINATION

The definition of systematic generalization (Definition 1) requires that each test label factor is seen in a training label. However, it is not directly used in the derivations, and the conclusions may apply to more general o.o.d. problems beyond recombining factors. A new output may correspond to an unseen activation for an output node, e.g., a new class in a classification problem. In such settings, it is sometimes discussed that the bias parameter of the output is a reason to avoid the new value prediction because it does not have any training signal to increase its value. This work provides another reason for not predicting a new value.

### 4.3 WHY DOES FUNCTION SHARING HAPPEN IN DEEP LEARNING?

We discuss that function sharing can be caused by characteristics of deep learning: deep architecture, shareable network, and greedy optimization. Deep architecture and shareable networks make it possible for factors to share elaborated functions, and greedy optimization encourages the sharing. Deep architecture and greedy search are necessary for deep learning. Deep learning uses deep architecture to fit complicated non-linear functions. Deep learning has a large and complex parameter space. To search in it, we need some prioritization, which leads to a greedy search. The shareable network is widely used in standard deep learning models and works well for i.i.d. problems. However, it is less critical compared to the other ones.

### 4.4 POTENTIAL SOLUTIONS

We consider possible solutions to avoid function sharing and achieve systematic generalization. From the above discussion, we look at shareable networks. We consider the recombination of factors and focus on sharing between factors. Then, one potential solution uses individual networks for output factors, similar to the experiment setup. We discuss how to design networks when the input or the output is entangled. If the output is entangled, we can design an architecture where each individual network only changes one factor in the output. For example, one individual network changes color, and another changes shape. If the input is entangled, we need to extract factors from it to feed the individual networks. It contains two questions: how to avoid spurious influence from other factors and keep it working in test distribution. We can bottleneck representations for the first one and divide the input into units invariant in training and test for the other, e.g., words or objects.

## 5 RELATED WORK

**Systematic generalization and deep learning**  Systematic generalization[2] (Fodor & Pylyshyn, 1988; Lake & Baroni, 2018; Bahdanau et al., 2019) is considered the "Great Move" of evolution, caused by the need to process an increasing amount and diversity of environmental information (Newell, 1990). Cognitive scientists see it as central for an organism to view the world (Gallistel & King, 2011). Studies indicate it is related to the prefrontal cortex (Robin & Holyoak, 1995). It was discussed that commonsense is critical (Mccarthy, 1959; Lenat et al., 1986) for systematic generalization, and recent works aim to find general prior knowledge (Goyal & Bengio, 2020), e.g.,

---

[2]It is also called compositional generalization in other literature.

Consciousness Prior (Bengio, 2017). Levels of systematicity were defined (Hadley, 1992; Niklasson & van Gelder, 1994), and types of tests were summarized (Hupkes et al., 2020). We focus on the primary case with an unseen combination of seen factor values.

A closely related field is causal learning, rooted in the eighteenth-century (Hume, 2003) and classical fields of AI (Pearl, 2003). It was mainly explored from statistical perspectives (Pearl, 2009; Peters et al., 2016; Greenland et al., 1999; Pearl, 2018) with do-calculus (Pearl, 1995; 2009) and interventions (Peters et al., 2016). The causation forms Independent Causal Mechanisms (ICMs) (Peters et al., 2017; Schölkopf et al., 2021). Systematic generalization is the counterfactual when the joint input distribution is intervened to have new values with zero probability in training (covariate shift). This work indicates that standard neural networks do not prefer to learn ICMs.

Parallel Distributed Processing (PDP) models (Rumelhart et al., 1986) use Connectionist models with distributed representations, which describe an object in terms of a set of factors. Though they have the potential to combine the factors to create unseen object representations (Hinton, 1990), it was criticized that they do not address systematic generalization in general (Fodor & Pylyshyn, 1988; Marcus, 1998). Deep learning is a recent PDP model with many achievements (LeCun et al., 2015; He et al., 2016). It was studied that deep neural networks use the composition of functions to achieve high performance (Montufar et al., 2014). The improvements in i.i.d. problems encourage to equip deep learning with systematic generalization.

**Recent directions**   In addition to architecture design (Russin et al., 2019; Andreas et al., 2016) and data augmentation (Andreas, 2020; Akyürek et al., 2021; Jia & Liang, 2016), the main perspectives for systematic generalization approaches include disentangled representation learning, attention mechanism, and meta-learning.

Disentangled representation (Bengio et al., 2013) is learned in unsupervised manners. Early methods learn the representation from statistical independence (Higgins et al., 2017; Locatello et al., 2019). Later, the definition of disentangled representation was proposed with symmetry transformation (Higgins et al., 2018). It leads to Symmetry-based Disentangled Representation Learning (Caselles-Dupré et al., 2019; Painter et al., 2020; Pfau et al., 2020). A disentangled representation learning model can be used as a feature extractor for other systematic generalization tasks.

Attention mechanisms are widely used in neural networks (Bahdanau et al., 2015). Transformers (Vaswani et al., 2017) are modern neural network architectures with self-attention. Recurrent Independent Mechanisms (Goyal et al., 2021b) use attention and the name of the incoming nodes for variable binding. Global workspace (Goyal et al., 2021a) improves them by using limited-capacity global communication to enable the exchangeability of knowledge. Discrete-valued communication bottleneck (Liu et al., 2021) further enhances systematic generalization ability.

Meta-learning (Lake, 2019) usually designs a series of training tasks for learning a meta-learner and uses it in a target task. Each task has training and test data, where test data requires systematic generalization from training data. When ICMs are available, they can be used to generate meta-learning tasks (Schölkopf et al., 2021). Meta-reinforcement learning was used for causal reasoning (Dasgupta et al., 2019). Meta-learning can also capture the adaptation speed to discover causal relations (Bengio et al., 2020; Ke et al., 2019).

Deep learning is a fast-growing field, and many efforts focus on designing architectures and algorithms to improve its performance. However, it is less discussed why standard deep learning models do not achieve systematic generalization. This paper looks into a built-in conflict.

## 6   CONCLUSION

This paper investigates a built-in conflict between deep learning and systematic generalization. It explains one of the reasons why standard neural networks seldom achieve systematic generalization. We hypothesize that the conflict is caused by sharing internal functions, and experiments support it. A model partitions an input space into multiple parts separated by boundaries. The function sharing tends to reuse the boundaries, leading to fewer parts for new outputs, which conflicts with systematic generalization. The phenomena are shown in different standard deep neural networks. We hope this finding provides a new understanding of systematic generalization mechanisms in deep learning and helps to improve machine learning algorithms for a higher level of artificial intelligence.

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

## A    MORE EXPERIMENTS

We run experiments with natural inputs. For image data, we use NICO++ dataset (Zhang et al., 2022). We use five foregrounds as the first output label and five backgrounds as the second. For text data, we use Amazon reviews (Ni et al., 2019). We use five categories as the first output label and five ratings as the second. Either dataset has two outputs, each containing five possible classes. There are 25 class combinations, and we separate them into 15 training combinations and 10 test ones in a similar way as in the experiment section. For fully connected networks, we use the Fashion dataset (ten classes) and render with ten colors. Please refer to Figure 6 and Table 1 for examples.

For the image dataset, we aggregated foregrounds into five abstract classes, e.g., mammal and vehicle. It better uses the limited data with annotations on combined labels. We use 72,176 image samples. For text data, we randomly select 100,000 samples for each category, with a length limit of 100 tokens. Data with training combinations are randomly split into training and i.i.d. generalization data with a ratio of 9:1.

The results are in Figure 7. Similar to the results in the experiment section, the test accuracies decrease as there are more shared layers. Also, both the training accuracy and the i.i.d. generalization accuracy do not decrease as much as the test accuracies.

We run ablations. Figure 8 shows the results with different layer widths. Figure 9 shows the results with dropout (Srivastava et al., 2014) and mixup (Zhang et al., 2018). All the results have similar effects as the original experiment.

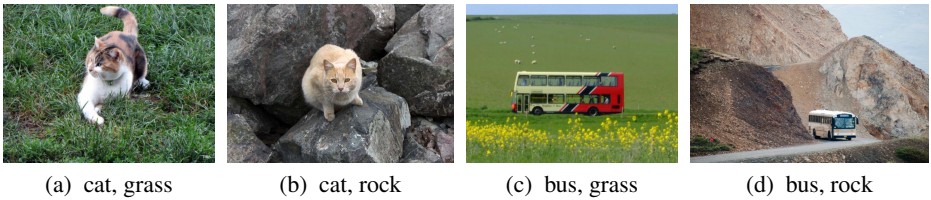

| (a)  cat, grass | (b)  cat, rock | (c)  bus, grass | (d)  bus, rock |

Figure 6: Examples of image data.

| Category | Rating | Review |
|---|---|---|
| Book | 5 | quick read from the most excellent author. fun |
| Book | 1 | It had tears, some label covering a defect and also wrinkled pages. |
| Electronics | 5 | Works perfectly, better than the original cable. |
| Electronics | 1 | DID NOT fit as described to accommodate the TV size! |

Table 1: Examples of text data.

## B    PROOFS

**Proposition 1** (Seen prediction). *From Assumption 2, $\forall x \in X : f(x) \in f(X_{train})$.*

*Proof.* We are going to prove that for a function $g$ with at least one input mapped to a new output, there exists a more preferred function $f$ with all inputs mapped to seen outputs. Therefore, if a function does not have other functions more preferred over it, the function follows the proposition.

Given a function $g$, we construct a function $f$. We pick a $x' \in X_{\text{train}}$. $\forall x \in X$:

$$g(x) \in g(X_{\text{train}}) : \quad f(x) = g(x),$$
$$\text{o.w.} : \quad f(x) = f(x').$$

Then, $\forall x_a, x_b \in X$ :

$$g(x_a) \in g(X_{\text{train}}) : \quad g(x_a) = g(x_b) \implies f(x_a) = g(x_a) = g(x_b) = f(x_b)$$
$$\text{o.w.} : \quad g(x_a) = g(x_b) \implies g(x_b) \notin g(X_{\text{train}}) \implies f(x_a) = f(x') = f(x_b)$$

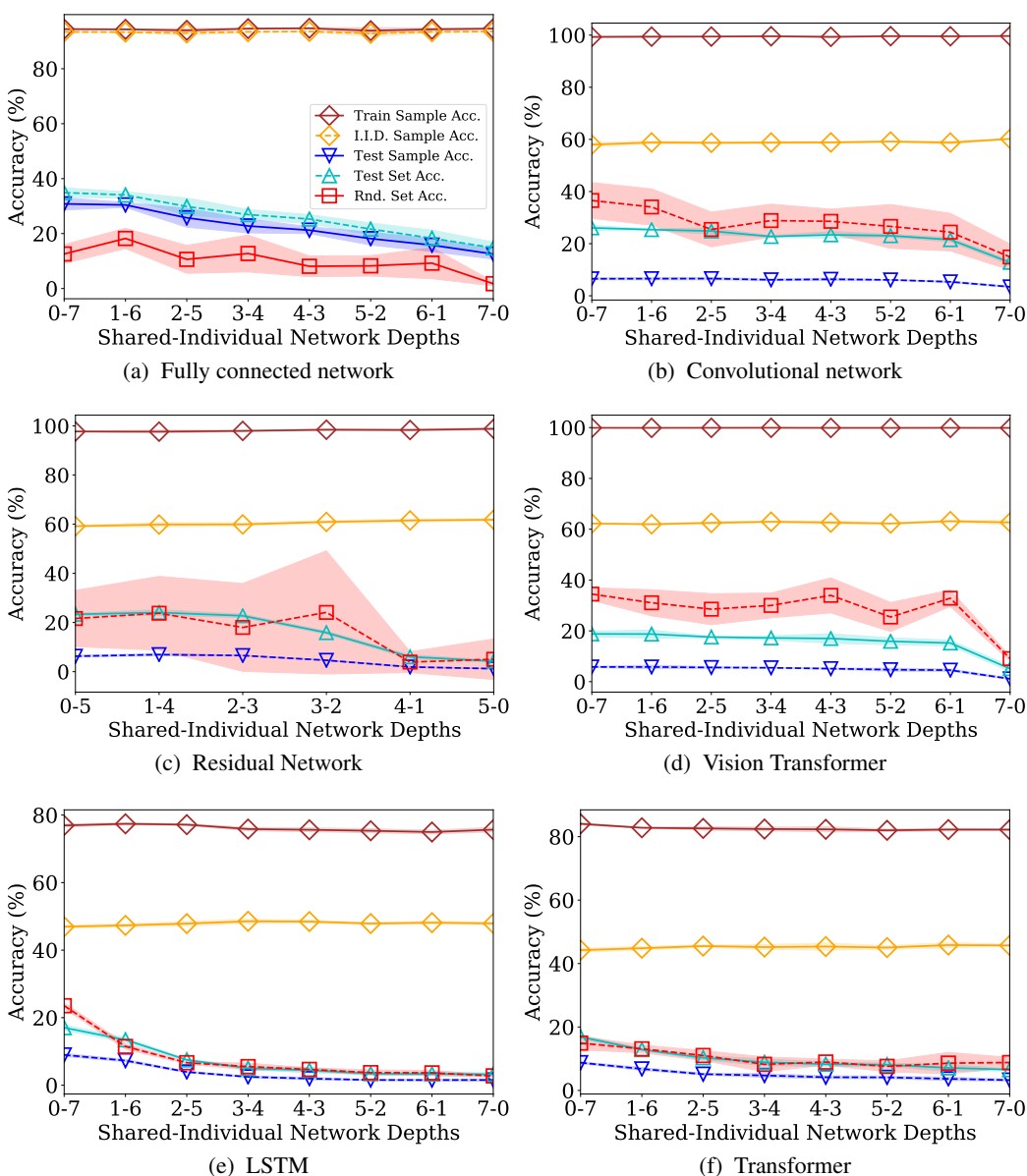

Figure 7: Results of three types of systematic generalization accuracy with shared and individual (sub)network depths. Original images from the NICO dataset and texts from Amazon reviews are used. Similar to Fignure 5, as the shared network depth increases, the accuracy generally decreases, indicating that function sharing reduces systematic generalization ability. Both the training and the i.i.d. generalization accuracy do not decrease as much as test accuracies.

In both cases,

$$g(x_a) = g(x_b) \implies f(x_a) = f(x_b)$$

On the other hand,

$$\exists x \in X_{\text{test}} : g(x) \notin g(X_{\text{train}}) \implies g(x) \neq g(x') \in g(X_{\text{train}}), \quad f(x) = f(x').$$

So $f(x) = f(x')$ does not imply $g(x) = g(x')$. With Assumption 2, $f$ is preferred over $g$.

Also, $\forall x \in X$:

$$g(x) \in g(X_{\text{train}}) : \quad \exists x'' \in X_{\text{train}} : g(x) = g(x'') \implies f(x) = f(x'') \in f(X_{\text{train}})$$
$$\text{o.w.} : \quad f(x) = f(x') \in f(X_{\text{train}})$$

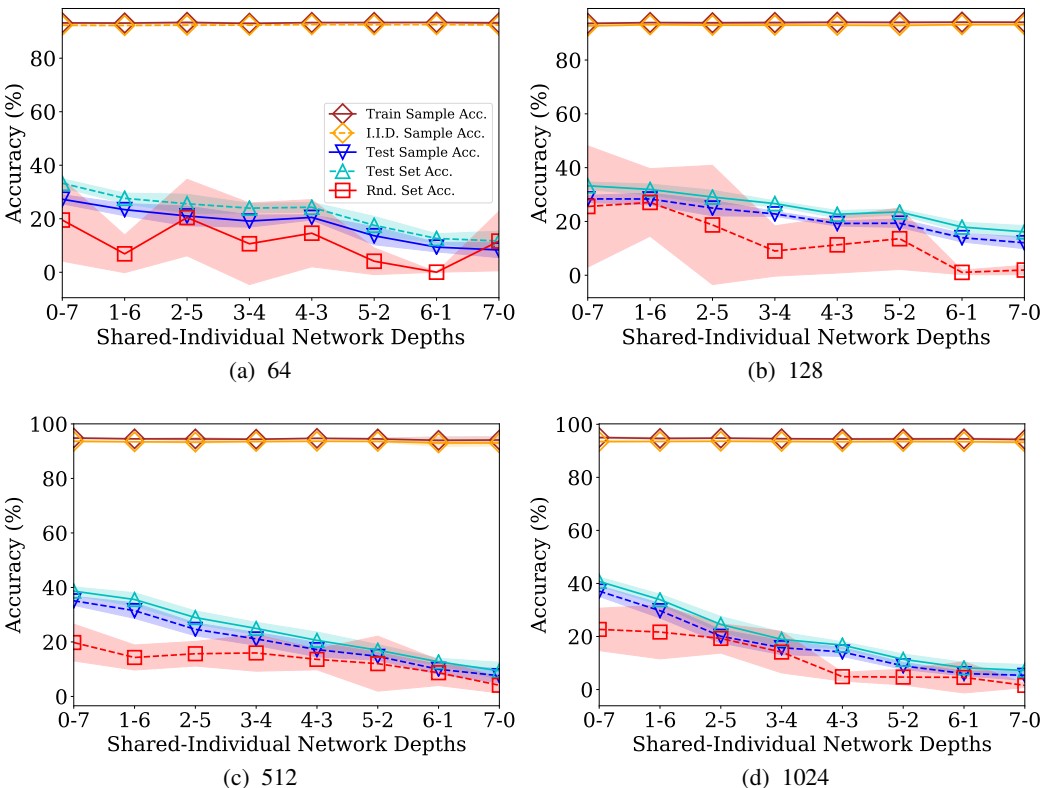

Figure 8: Layer width ablation results of three types of systematic generalization accuracy with shared and individual (sub)network depths. Similar to Fignure 7a (width is 256), as the shared network depth increases, the accuracy generally decreases, indicating that function sharing reduces systematic generalization ability. Both the training and the i.i.d. generalization accuracy do not decrease as much as test accuracies.

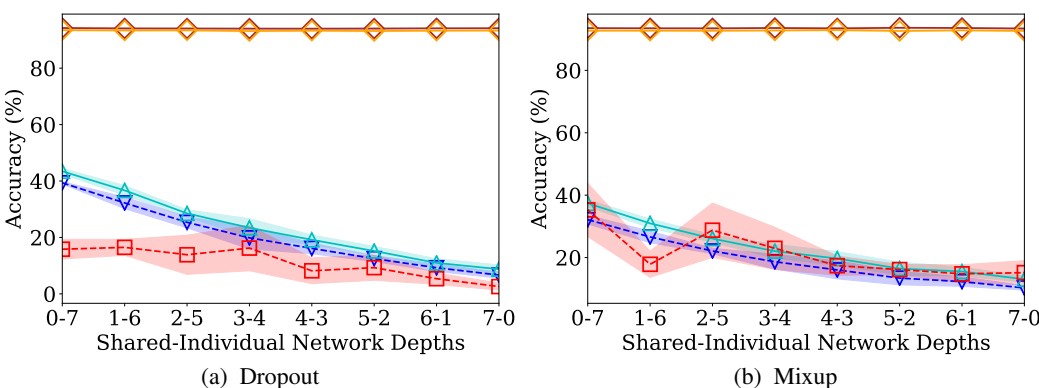

Figure 9: Dropout and mixup ablation results of three types of systematic generalization accuracy with shared and individual (sub)network depths. Similar to Fignure 7a, as the shared network depth increases, the accuracy generally decreases, indicating that function sharing reduces systematic generalization ability. Both the training and the i.i.d. generalization accuracy do not decrease as much as test accuracies.

Therefore, $\forall x \in X : f(x) \in f(X_{\text{train}})$. □

**Proposition 2** (Seen label). *From Assumption 1 and Proposition 1, $\forall x \in X : f(x) \in Y_{train}$.*

*Proof.* From Assumption 1, $f(X_{\text{train}}) = Y_{\text{train}}$. From Proposition 1,

$$\forall x \in X : f(x) \in f(X_{\text{train}}) = Y_{\text{train}}$$

$\square$

**Theorem 1** (The conflict). *From Definition 1 and Proposition 2, $\forall(x,y) \in \mathcal{D}_{test} : y \neq f(x)$.*

*Proof.* $\forall(x,y) \in \mathcal{D}_{\text{test}}$: $y \notin Y_{\text{train}}$ (Definition 1), $x \in X_{\text{test}} \subseteq X \implies f(x) \in Y_{\text{train}}$ (Proposition 2). Therefore $y \neq f(x)$. $\square$

## C  A CONJECTURE ON WHY THE FUNCTION SHARING HAPPENS

The neural network training process is complicated, so it is hard to describe what happens. Instead, we have a conjecture.

*For a large network, when a boundary is learned for the first time, it separates the problem into two sub-problems, and their learning processes do not influence each other for the rest of the training.*

We first explain the idea with an analogy of a binary decision tree (or a hierarchical classification). We then define a boundary and discuss its properties.

**Binary Decision tree**    We consider such a binary decision tree that at each decision node, a label set is separated into two parts, each for a sub-tree. For example, if there are 10 classes, the root node may divide them into 6 and 4 classes. Then the root node of the first sub-tree divides the 6 lasses into two sets of 3 classes. In this decision tree, a node separates input space into two parts with disjoint output labels, and each part is learned separately.

Such a decision tree does not predict new outputs because all leaf nodes predict seen outputs. We discuss neural network training process is similar to creating such a decision tree from some aspects. A decision tree node is a boundary in a neural network.

**Boundary**    We consider a problem $P = (\mathcal{X}, X, Y)$ with input space $\mathcal{X}$, input set $X$, output set $Y$. We have ground truth mapping $f : X \to Y$ and learned mapping $\hat{f} : \mathcal{X} \to Y$. We define a boundary as follows.

**Definition 2** (Boundary). *Suppose we have a binary partition $(\mathcal{X}_a, \mathcal{X}_b)$ of an input space $\mathcal{X}$. $\mathcal{X}_a, \mathcal{X}_b$ are non-empty.*

$$\mathcal{X}_a \dot{\cup} \mathcal{X}_b = \mathcal{X}, \quad X_a = X \cap \mathcal{X}_a, \quad X_b = X \cap \mathcal{X}_b, \quad Y_a = f(X_a), \quad Y_b = f(X_b).$$

*It is a boundary if the output sets are disjoint; for each part, all inputs map to its output set.*

$$Y_a \dot{\cup} Y_b = Y, \quad \hat{f}(\mathcal{X}_a) = Y_a, \quad \hat{f}(\mathcal{X}_b) = Y_b.$$

A boundary separates a problem $P = (\mathcal{X}, X, Y)$ to two sub-problems $P_a = (\mathcal{X}_a, X_a, Y_a)$ and $P_b = (\mathcal{X}_b, X_b, Y_b)$. We assume that when the boundary is learned for the first time, the learning processes of $P_a$ and $P_b$ do not influence each other for the rest of the training.

**Assumption 3** (Separate sub-problems). *When a network is large enough, learning one sub-problem does not affect the prediction of another sub-problem.*

With this assumption, for a large network, a boundary separates the original problem into two problems whose training processes do not influence each other. The assumption applies to each sub-problem. When a problem has only one label, all the inputs are mapped to the label. So the model is learned not to predict an unseen output. This learning process is similar to that of a decision tree.

## D  EXPERIMENT DETAILS

### D.1  VISUALIZATION SETTINGS

The model is a fully connected neural network with two input and two output nodes. It has six hidden layers with ReLU activations, and each hidden layer has eight nodes. We use a mini-batch

size of 10 with a learning rate of 0.01. We iterate until the model prediction becomes stable. Please see the original work of deep playground for more information. We use six Intel(R) Core(TM) i5-8400 2.80GHz CPUs, and the asset has a public license.

## D.2 EXPERIMENT SETTINGS

We use GeForce GTX 1080 or GeForce GTX 1050 Ti GPU for single GPU experiments. We use TensorFlow for implementation. The assets have a public license.

Each input element is linearly scaled to [-0.5, 0.5] for image input. We also uniformly sample from this interval for random image input. We select two sentence lengths uniformly from valid integers (one to maximum length) and then generate each word uniformly from the vocabulary for random text input.

**Fully Connected Network**   The input shape is $28 \times 28$, flattened to a vector. There are seven fully connected layers. Each of them has 512 hidden nodes and ReLU activation. The output has ten nodes and Softmax activation. We use cross-entropy loss and Adam optimizer with a learning rate of 0.001. The batch size is 512, and we train 2,000 iterations. Each evaluation uses 10,000 samples.

**Convolutional Network**   The input shape is $32 \times 32 \times 3$. There are seven convolutional layers. Each of them has $3 \times 3$ kernel size with 64 channels. Then the layer is flattened. We have a fully connected layer with 128 nodes and ReLU activation. The output layer has ten nodes with Softmax activation. We use cross-entropy loss and Adam optimizer with a learning rate of 0.001. The batch size is 512, and we train 5,000 iterations. Each evaluation uses 10,000 samples.

**Residual Network**   The input is the same as CNN. The model is the standard ResNet50 implementation. The hidden groups are treated as one layer, so there are five hidden layers. The hidden layer size is 64. The output layer has ten nodes with Softmax activation. We use cross-entropy loss and Adam optimizer with a learning rate of 0.001. The batch size is 512, and we train 10,000 iterations. Each evaluation uses 10,000 samples.

**Vision Transformer**   The input is the same as CNN. The model is the standard Vision Transformer implementation with seven hidden layers. The hidden layer size is 64. The output layer has ten nodes with Softmax activation. We use cross-entropy loss and Adam optimizer with a learning rate of 0.001. The batch size is 512, and we train 10,000 iterations. Each evaluation uses 10,000 samples.

**LSTM**   The vocabulary size is 30,977, including a start symbol and padding symbol. The input length is 200. The embedding size is 64. There are seven stacked bidirectional LSTM layers, and each has 32 hidden nodes for each direction. Then the output is flattened. The output layer is a fully-connected layer with ten output nodes and Softmax activation. We use cross-entropy loss and Adam optimizer with a learning rate of 0.001. The batch size is 64, and we train 1,000 iterations. Each evaluation uses 10,000 samples.

**Transformer**   The input is the same as that of LSTM. The embedding size is 64. There are seven hidden groups. The hidden layer size is 64. The output is flattened. The output layer is a fully-connected layer with ten output nodes and Softmax activation. We use cross-entropy loss and Adam optimizer with a learning rate of 0.001. The batch size is 64, and we train 2,000 iterations. Each evaluation uses 10,000 samples.

## D.3 EXPERIMENT RESULTS

We numerically compare the individual (left ends in result figures, 0-max) and shared (right ends, max-0) networks in Table 2. LSTM is the stacked LSTM, and LSTM-1 has only one LSTM layer. It shows that shared network has lower scores than individual network on the three types of accuracy. So it indicates that function sharing avoids systematic generalization.

Table 2: Accuracy (mean $\pm$ std %). Experiment results.

| | Test Sample Accuracy | | Test Set Accuracy | | Random Set Accuracy | |
| | Individual | Shared | Individual | Shared | Individual | Shared |
|---|---|---|---|---|---|---|
| DNN | $41.6 \pm 1.0$ | $9.2 \pm 0.7$ | $47.1 \pm 1.4$ | $10.2 \pm 0.6$ | $36.9 \pm 0.8$ | $7.1 \pm 0.6$ |
| CNN | $41.6 \pm 0.4$ | $21.3 \pm 0.8$ | $61.3 \pm 1.0$ | $30.6 \pm 1.0$ | $81.4 \pm 4.0$ | $41.4 \pm 13.7$ |
| ResNet | $41.8 \pm 0.8$ | $8.3 \pm 1.6$ | $59.4 \pm 1.4$ | $11.8 \pm 1.9$ | $74.1 \pm 9.0$ | $17.9 \pm 8.5$ |
| ViT | $29.3 \pm 0.9$ | $3.8 \pm 0.4$ | $40.2 \pm 0.9$ | $5.6 \pm 0.5$ | $79.2 \pm 3.0$ | $27.9 \pm 6.1$ |
| LSTM | $26.0 \pm 1.5$ | $21.6 \pm 2.2$ | $41.1 \pm 1.2$ | $29.2 \pm 2.3$ | $16.0 \pm 5.9$ | $5.0 \pm 2.7$ |
| LSTM-1 | $28.6 \pm 2.2$ | $26.0 \pm 1.9$ | $40.1 \pm 2.0$ | $34.8 \pm 2.1$ | $15.1 \pm 4.1$ | $8.0 \pm 2.1$ |
| Transformer | $28.4 \pm 2.5$ | $22.1 \pm 2.1$ | $48.6 \pm 1.7$ | $33.9 \pm 3.2$ | $45.1 \pm 8.0$ | $16.4 \pm 8.4$ |

## E  ZERO-SHOT LEARNING DATASETS

We look at the results for Zero-shot learning. We use aPY (Farhadi et al., 2009), AwA2 (Xian et al., 2019), CUB (Wah et al., 2011), and SUN (Patterson & Hays, 2012) datasets. In these datasets, factors are mainly related to the input locality (Sylvain et al., 2020). We use the pre-extracted input features for aPY and AwA and image input for CUB and SUN. We construct output labels from attributes. Each attribute is a binary number, 1 for existing in a sample and 0 otherwise. We select six attributes with the most balanced average number (close to 0.5) in all data. The first, third, and fifth attributes are used for the first output, and the others for the second output. Each output has eight classes yielded from all the combinations of three binary attributes. For aPY, samples may share the same image, so we construct training data from the original training and test data from the original test data. For other datasets, we split all data to disjoint training and test data.

We use an eight-layer CNN model, the same as in the experiment section. The batch size is 512 for aPY and AwA and 256 for CUB and SUN. Other settings are the same as that in the experiment section. The result is shown in Figure 10 and Table 4. Similar to the previous experiments, the depths of shared and individual networks reduce systematic generalization capability.

Table 3: Accuracy (mean $\pm$ std %). Experiment results for Zero-Shot Learning.

| | Test Sample Accuracy | | Test Set Accuracy | | Random Set Accuracy | |
| | Individual | Shared | Individual | Shared | Individual | Shared |
|---|---|---|---|---|---|---|
| aPY | $2.5 \pm 0.3$ | $0.6 \pm 0.2$ | $22.4 \pm 2.3$ | $5.9 \pm 1.2$ | $36.6 \pm 30.6$ | $6.8 \pm 2.8$ |
| AwA2 | $1.6 \pm 0.4$ | $0.6 \pm 0.1$ | $21.2 \pm 3.0$ | $9.9 \pm 1.0$ | $45.2 \pm 3.8$ | $18.5 \pm 0.3$ |
| CUB | $2.5 \pm 0.2$ | $1.2 \pm 0.2$ | $39.7 \pm 0.4$ | $22.2 \pm 1.2$ | $38.8 \pm 2.6$ | $21.9 \pm 3.7$ |
| SUN | $1.9 \pm 0.1$ | $1.1 \pm 0.3$ | $15.3 \pm 1.0$ | $8.5 \pm 0.5$ | $35.9 \pm 11.7$ | $8.3 \pm 4.2$ |

## F  MORE DISCUSSIONS

### F.1  TRAINING PROCESS

We discussed that sharing boundaries reduces the number of partitions and shrinks the area for new outputs (Proposition 1). We run experiments to find when this happens during training. We sample 10,000 inputs from test data, and if an output combination has at least 50 samples, we regard it as a new output (o.o.d.) partition. We plot the number of o.o.d. partitions and the test sample ratio in the o.o.d. partitions for shared and individual networks in Figure 11. The experiment settings follow DNN and CNN settings in the experiment section. The results show that the differences start to happen in early training.

### F.2  EQUALLY DIFFICULT FACTORS

We look at the results when two inputs are equally hard to learn. We use two CIFAR-10 datasets for both the first and the second datasets. Since the average of two colored images can cause training data under-fitting, we merge the inputs by concatenating them by channel. It means the input channel is six. The result is shown in Figure 12 and Table 4. Similar to the previous experiments, when the difficulties are equal, the depth of the shared network weakens systematic generalization.

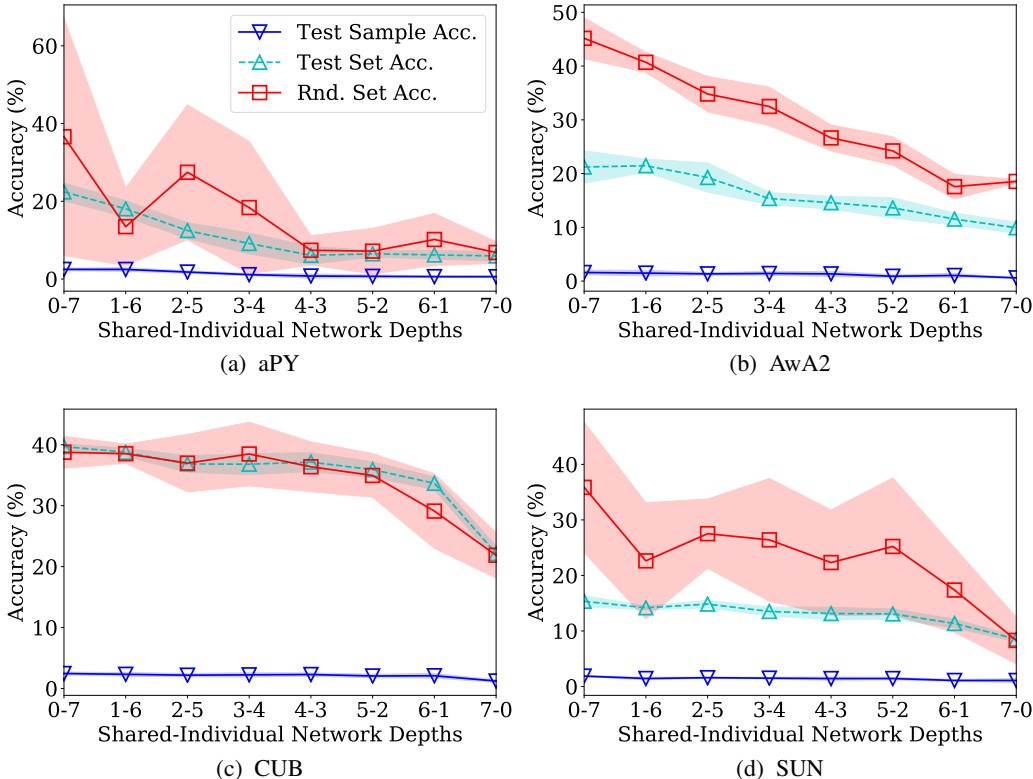

Figure 10: Experiment results for Zero-Shot Learning with CNN models. The results are similar to those in the experiment section.

Table 4: Accuracy (mean ± std %). Experiment results for equally difficult factors.

|  | Test Sample Accuracy | | Test Set Accuracy | | Random Set Accuracy | |
|---|---|---|---|---|---|---|
|  | Individual | Shared | Individual | Shared | Individual | Shared |
| DNN | 24.6 ± 0.9 | 4.4 ± 0.5 | 33.3 ± 1.3 | 6.4 ± 0.6 | 31.2 ± 2.5 | 5.4 ± 0.6 |
| CNN | 52.5 ± 1.5 | 39.6 ± 1.2 | 60.3 ± 1.6 | 45.2 ± 1.5 | 32.4 ± 3.2 | 20.5 ± 4.7 |

### F.3 LABEL COMBINATIONS

We also test other training label distribution types. We design tile and one-shot combinations. In tile, a label combination is for training when $Y_1 < 5$ or $Y_2 < 5$. It is similar to the split for illustrative example. In One-shot, a label combination is for training when $Y_1 < 9$ or $Y_2 < 1$. In such a case, only $(9,0)$ contains $Y_2 = 0$. It is similar to one-shot learning. The results for the fully connected neural network are in Figure 13 and Table 5. It is similar to the results in the experiment section.

Table 5: Accuracy (mean ± std %). Experiment results for other label combinations.

|  | Test Sample Accuracy | | Test Set Accuracy | | Random Set Accuracy | |
|---|---|---|---|---|---|---|
|  | Individual | Shared | Individual | Shared | Individual | Shared |
| Tile | 33.0 ± 3.9 | 8.3 ± 1.1 | 36.5 ± 4.6 | 9.2 ± 1.1 | 9.7 ± 2.5 | 1.5 ± 0.1 |
| One-shot | 20.9 ± 13.3 | 1.4 ± 0.8 | 23.1 ± 14.9 | 1.5 ± 0.8 | 0.0 ± 0.0 | 0.0 ± 0.0 |

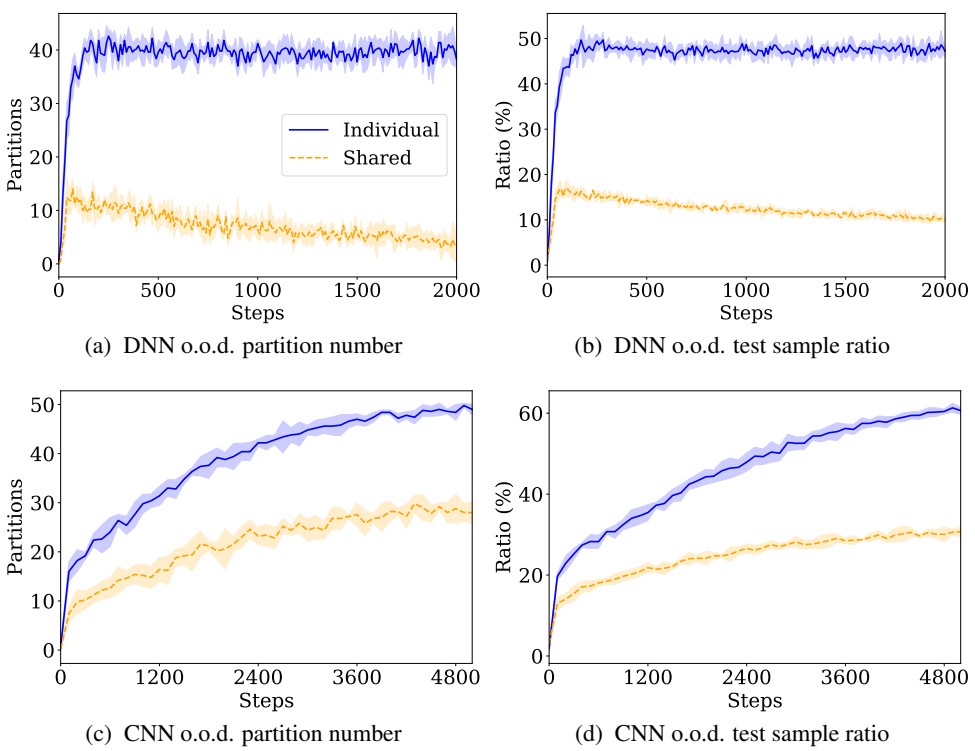

Figure 11: Training process. The o.o.d. partition number and the o.o.d. test sample ratio for shared and individual networks.

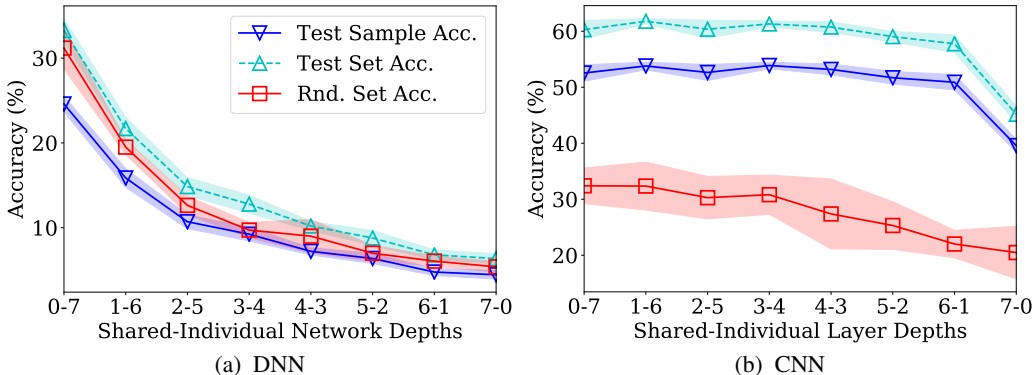

Figure 12: Experiment results for equally difficult factors. The results are similar to those in the experiment section.

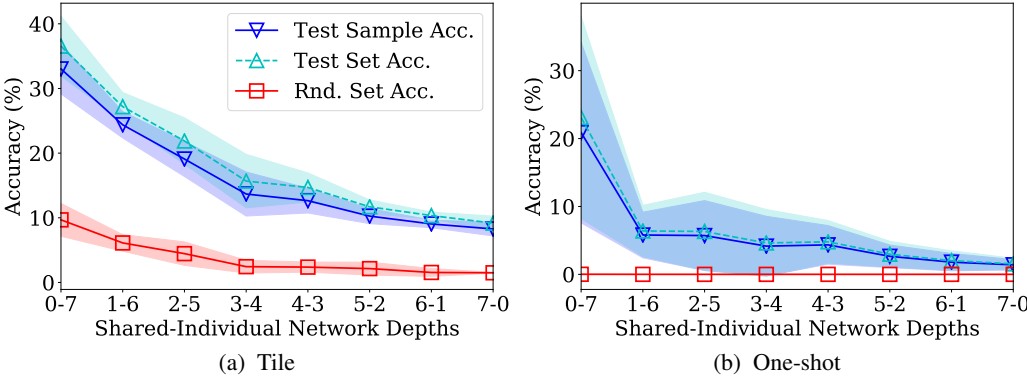

Figure 13: Experiment results for other label combinations. The results are similar to those in the experiment section.

