# OpenReview forum: "On a Built-in Conflict between Deep Learning and Systematic Generalization"
_ICLR.cc/2023/Conference — Submitted to ICLR 2023_

### Official Review · Reviewer_4tfA · 2022-10-12

**Confidence:** 4
**Correctness:** 3
**Technical Novelty And Significance:** 3
**Empirical Novelty And Significance:** 2
**Recommendation:** 5

**Clarity, Quality, Novelty And Reproducibility:**

Clarity: the general idea of the paper is easy to understand but the details are not explained in a very clear way.
Novelty: as far as I know, the hypothesis introduced in this paper is new.

**Strength And Weaknesses:**

Strengths: the hypothesis advanced by this paper is new and interesting; it deserves to be developed further.

Weaknesses:
- The paper is not very well written; see below for some examples where the writing should be improved.
- The propositions and theorem are trivial.
- The vision experiments seem quite unnatural, since images from different datasets are averaged (see Section 3.2). Also the text experiments are not particularly natural (they concatenate two unrelated sequences and ask to predict both labels).

A general question that might be worth addressing: the shape of the regions is not taken into account? For instance, to prove Proposition 1 (using Assumption 2) you can simply merge regions, but it might be unnatural to merge them, given the geometry.

Examples of unclear parts in Section 2.1:
- "$Y$ contains $K$ factors $Y_1$, . . . , $Y_K$" is not clear. Does it mean that $Y = Y_1 \times \dots \times Y_K$? But this would be at odds with the (unclear) statement "which can be entangled".
- Is $X_\text{train}$ the (ordered) sequence of inputs or is it a set that contains all the input data? (same for $Y_\text{train}$ and test)
- "The values for each factor $i$ are included in the training output" is also unclear
- "A model $f$ maps input $X$ to the prediction of output $f (X)$": so is $X$ the set of all inputs or a single input?

Examples of unclear parts in Section 2.2:
- "deep learning more or equally prefers $f$ over $g$" -> "deep learning prefers $f$ over $g$ more or equally" (a bit more clear in my opinion)

Examples of unclear parts in Section 3.1:
- "$Y_1$ is chosen from all possible labels": what does this mean? That $Y_1$ is the set of all labels?
- The two datasets share the same input space $X$?



**Summary Of The Paper:**

This paper hypothesizes that systematic generalization is fundamentally at odds with the tendency of deep learning models to share sub-modules. It provides both a theoretical description and derivation of the issue as well as supporting experiments with several deep learning architectures.

**Summary Of The Review:**

This paper introduces an interesting and probably novel idea but it does not a good job at developing it. I believe it does not meet the ICLR bar.

---

> ### Author Response · Authors · 2022-11-19
> **Thank you for the comments.**
>
> Q1: The shape of the regions is not taken into account.
>
> A1: That is right. The geometry can be complicated and disconnected for data distribution and model predictions. So we do not consider the shape of regions.

---

> > ### Comment · Reviewer_4tfA · 2022-11-21
> > **Thank you for the revision**
> >
> > I think the new "natural" experiments are useful and they should constitute the bulk of the experimental section in the main paper. (I understand that the time available for the rebuttal is probably not enough for a large refactoring.)
> >
> > Overall, I believe the weaknesses described in my original review mostly remain in the current revision, so I will keep my score.

---

### Official Review · Reviewer_hir1 · 2022-10-20

**Confidence:** 5
**Correctness:** 2
**Technical Novelty And Significance:** 2
**Empirical Novelty And Significance:** 3
**Recommendation:** 5

**Clarity, Quality, Novelty And Reproducibility:**

The clarity of the paper could be improved. For example:
* Showing examples of the task stimuli in the paper—particularly the visual ones—would I think help to emphasize how unnatural the tasks are.
* The data preparation could be rewritten to be clearer, by first specifying that the data generating process goes from sampling a pair of labels to sampling the corresponding input.
* Split architecture details were unclear to me (noted above).

There is some originality and quality if the above weaknesses are addressed.

**Strength And Weaknesses:**

Strengths:
* The questions posed are interesting.
* I appreciate the breadth of architectures considered, especially sharing different numbers of layers.


Weaknesses:

The way the paper is presented fundamentally ignores the No Free Lunch theorem [e.g. Adam, 2019]. There is no system that can generalize perfectly on every task and training dataset—there cannot be a conflict between an architecture class and systematic generalization writ large. We have to ask the question of how the inductive biases of the model class fit the class of tasks we are interested in solving.
* DL researchers are well aware of the feature sharing bias—it forms the basis of auxiliary task training and/or pretraining methods, as the authors note. The reason such methods tend to improve generalization is because sharing features is useful on real-world datasets. For example, the input features learned solve masked language modeling tasks empirically improve systematic generalization performance substantially even on tasks like SCAN and CFQ [Furrer et al., 2020]. There are even theoretical accounts of why feature sharing can improve generalization in the presence of noise [e.g. Lampinen et al, 2019].
* The datasets used in the paper are therefore cleverly created to make feature sharing an actively harmful strategy. But to do so, the authors rely on essentially adversarial dataset design, where they combine input stimuli in very unnatural ways (averaging images, or concatenating completely unrelated pieces of text), and then enforce extremely strong correlations between these inputs at train time, which are completely reversed at test time. There is no reason given to think that this process has anything to do with any real-world data generating process.
* Therefore, I would challenge the authors to demonstrate *real-world tasks and datasets*, not artificially, adversarially created ones, in which their observations apply.
* Otherwise, it seems to me that feature sharing is a *feature, not a bug* of deep learning. Nobody has ever claimed that deep learning is capable of generalizing systematically in every task anyone can come up with—that would violate the NFL theorem. But I’d argue that the DL family is empirically the most successful system for generalizing on real world datasets, in part because of feature sharing.

Architectures and training paradigms:
* “We choose a layer and duplicate the following layers, keeping the number of all hidden nodes in each layer if feasible” — it is not clear to me whether this means that each “branch” of the architecture has the same number of nodes as before, or half the number of nodes. If the former, the number of parameters will be larger in networks that split earlier, thus confounding the comparison (since overparameterized models tend to generalize better).
* More generally, it would be interesting to see the impact of parameterization on these effects—one might expect somewhat less feature sharing in wider networks, for instance, though it’s unclear how strong the effect would be.
* And it would be interesting to see the effect of methods like dropout [Srivastava et al., 2014] or mixup [Zhang et al., 2017] which are known to improve generalization.


References
------------

Adam, Stavros P., et al. "No free lunch theorem: A review." Approximation and optimization (2019): 57-82.


Furrer, D., van Zee, M., Scales, N., & Schärli, N. (2020). Compositional generalization in semantic parsing: Pre-training vs. specialized architectures. arXiv preprint arXiv:2007.08970.

Lampinen, A. K., & Ganguli, S. (2019). An analytic theory of generalization dynamics and transfer learning in deep linear networks. In International Conference on Learning Representations.

Srivastava, Nitish, et al. "Dropout: a simple way to prevent neural networks from overfitting." The journal of machine learning research 15.1 (2014): 1929-1958.

Zhang, H., Cisse, M., Dauphin, Y. N., & Lopez-Paz, D. (2017). mixup: Beyond empirical risk minimization. arXiv preprint arXiv:1710.09412.


**Summary Of The Paper:**

This paper investigates systematic generalization of multi-label classifications in settings where the input space is shared between the different labels. The authors suggest that deep learning models are biased towards feature reuse, which conflicts with systematic generalization to new combinations of known classes. They explore this hypothesis with theoretical analyses under toy assumptions, and empirical experiments across a variety of architectures. Generally, they find that more sharing of features reduces generalization accuracy in their settings.

**Summary Of The Review:**

See comment above for my post-response update. I am updating my score in accordance with the improvement in the paper, but I am concerned that key issues still remain unresolved, and not discussed with enough nuance.

Original review
-------------------

If this paper were completely rewritten—to describe the experiments as identifying a particular class of problems in which DL architectures seem not to generalize systematically due to feature-sharing rather than a built-in conflict—I believe it could be an acceptable paper in some venue. If, in addition to that, the authors were to identify real-world, non-adversarial datasets where their observations bear out, and perform some of the architecture/training experiments suggested above, I would consider it a strong paper for NeurIPS. As it is, I think it is misleading.

---

> ### Author Response · Authors · 2022-11-19
> **Thank you for the comments.**
>
> Q1: “We choose a layer and duplicate the following layers, keeping the number of all hidden nodes in each layer if feasible” — it is not clear to me whether this means that each “branch” of the architecture has the same number of nodes as before, or half the number of nodes.
>
> A1: It has half number of nodes. (Vision) transformers require all the layers has the same size, so the brach have the same number of nodes as before in these models.
>
> Q2: If the former, the number of parameters will be larger in networks that split earlier, thus confounding the comparison (since overparameterized models tend to generalize better).
>
> A2: We report i.i.d. generalization results in the additional experiments and their changes are smaller than the changes of systematic generalization results.

---

### Official Review · Reviewer_e3dk · 2022-10-26

**Confidence:** 3
**Correctness:** 2
**Technical Novelty And Significance:** 2
**Empirical Novelty And Significance:** 3
**Recommendation:** 5

**Clarity, Quality, Novelty And Reproducibility:**

As mentioned above, there are some significant issues with clarity, but I believe it is possible for them to be resolved in an updated version of the paper. The paper also seems novel enough in that it looks at function sharing as a potential underlying cause for lack of systematic generalization. The quality is below average, with issues including incomplete experimental evaluation and unsupported claims (e.g. greedy learning of functions and the mechanisms underlying function sharing). I also found the design choice to average inputs from two separate datasets to be unconventional.

One point that seems quite relevant but not addressed by the paper is the extent to which the phenomenon in Figure 2 is caused by a softmax activation, which assumes that classes are mutually exclusive. This seems to be an alternate reason that the case in Figure 2(b) does not arise, since this region occupied by the orange dot would be a region of low confidence and thus the network would be incentivized to sharpen the decision boundaries. Could this possibly be resolved by assuming a multi-output loss function, e.g. multiple sigmoids? Regarding reproducibility, code is included and thus reproducing the results does not seem to be a major barrier.

**Strength And Weaknesses:**

Strengths
- The broad aim of investigating stronger forms of generalization that move beyond the i.i.d. case is interesting.
- The experimental observation that having fewer shared layers leads to better systematic generalization holds across multiple diverse architectures and datasets.
- As far as I know, investigating function sharing as a reason for lack of systematic generalization is a novel approach.

Weaknesses
- The paper is not written very clearly. Sections that are difficult to understand include: the mathematical notation (e.g. writing that $f$ is a "model" but not explaining that this is simply a mapping from the input space to output space), the experiment section (particularly how the labels were generated and what the different evaluation metrics mean), and the discussion section.
- Some intuitions are claimed but not supported by adequate evidence: that deep neural networks prefer to learn a simple function and combine with previously learned functions, and that neural networks greedily learn functions in order of simpler to more complex.
- The experiments are not complete. Some relevant but missing pieces of information include training accuracy and computation time for the various levels of sharing. On a related note, the motivation for including the test set and random set accuracy metrics is unclear.
- It is not clear how the knowledge introduced by this paper can be effectively used to improve systematic generalization. Training a multitude of independent networks, one for each underlying factor, does not seem like a practical course of action due to storage and computation constraints.

**Summary Of The Paper:**

This paper investigates systematic generalization in deep neural networks. Systematic generalization here refers to the ability of an algorithm to produce outputs that were not observed during training time. A potential reason for this is postulated: the lack of systematic generalization in deep neural networks is due to function sharing, i.e. that each layer in the network uses a common representation from the previous layer. Experiments show that networks with fewer shared intermediate layers exhibit a greater degree systematic generalization than those with more shared layers.

**Summary Of The Review:**

After rebuttal: The paper has been improved by the inclusion of real-world experiments. I would echo the other reviewers in suggesting that these move to the main paper. As pointed out by Reviewer hir1, a more complete discussion about when function sharing is likely to be helpful seems necessary. Expanding on why function sharing occurs as started in Appendix C would also be useful. I have updated my score accordingly.

---

Overall, there are some potentially interesting ideas in this paper, but the clarity, unsupported claims, and incompleteness of the experiments are somewhat significant issues. The contribution of this paper is on the more incremental side and consists primarily of showing that sharing fewer layers can improve systematic generalization. However, it is not clear how to take these insights and apply them to solve out-of-distribution detection on real-world problems.

---

> ### Author Response · Authors · 2022-11-19
> **Thank you for the comments.**
>
> Q1: One point that seems quite relevant but not addressed by the paper is the extent to which the phenomenon in Figure 2 is caused by a softmax activation, which assumes that classes are mutually exclusive. This seems to be an alternate reason that the case in Figure 2(b) does not arise, since this region occupied by the orange dot would be a region of low confidence and thus the network would be incentivized to sharpen the decision boundaries.
>
> A1: It seems the sharpness of the decision boundaries might not directly influence the sharing. For example, in Figure3, the top right corner has a not sharp boundary, but it is still shared.
>
> The information of a boundary is that the two-side of it have different values. The difference can be between two one-hot vectors or between two multi-hot vectors. So it seems that using a multi-output loss function is not directly related.

---

### Official Review · Reviewer_w8Ad · 2022-10-30

**Confidence:** 4
**Correctness:** 3
**Technical Novelty And Significance:** 2
**Empirical Novelty And Significance:** 2
**Recommendation:** 3

**Clarity, Quality, Novelty And Reproducibility:**

In general, the writing needs to be improved. It was not easy to follow all the details:
- From the start, it was difficult to understand what a function means in deep learning context? To understand it better w.r.t machine learning literature, can “function sharing” be reframed in terms of modularity?
- Figure 2 is hard to understand. Eg. what does the term “function” refers to in the diagram?

**Strength And Weaknesses:**

### Strengths
- The paper provides empirical evidence of why parameter sharing (function sharing) leads to performance drop in systematic generalization for deep learning models.
- I like the problem space and believe that the authors are on to something tangible here but the lack of rigour in analysis didn't convince me.

### Weaknesses
- I have issues with problem formulation and writing. Some details of the work are not framed correctly and it’s hard to understand it since no context from previous literature is provided while introducing and explaining new concepts.
- Eg. what is implied as functions in deep neural networks? Is it an individual parameter or a set of parameters?
- What does the three equations at the end of sec 3.1 refer to?
- There is no clear description of the dataset. I can see that it contains factor but what are those factors? How are those factors combined?
- On the same note, could you please provide description of the datasets and models used, separately?
- The results section just provide information on the models tried. No information on training, and test, train splits of dataset provided.
- The author try to ablate different model architectures, however they use different datasets across those architectures thus it’s hard judge if the results are consistent across those architectures or the differences arise from the difference in datasets.


**Summary Of The Paper:**

The paper hypothesise that function sharing is one of the reasons why deep learning models can’t perform systematic generalization. The paper demonstrate that as the degree of parameter sharing increases, the systematic generalization drops. From the practical stand point, the papers argues for sparsity in models that somewhat learn symbolic functions (in term of disentangling feature attributes). Although it’s not touched upon but I believe the paper can be seen from modularity perspective where each module encampasses a particular underlying function, describing certain factor of the input.

**Summary Of The Review:**

This work need major re-writing as most of the concepts were not framed correctly. Moreover, they are some shortcomings in the evaluation section as I explained in the weaknesses section. I believe there is some value in this work, but it needs written with clarity.

---

> ### Author Response · Authors · 2022-11-19
> **Thank you for the comments.**
>
> Q1: What does the three equations at the end of sec 3.1 refer to?
>
> A1: They refer to the three accuracy metrics defined in the paragraph above.
>
> Q2: The author try to ablate different model architectures, however they use different datasets across those architectures thus it’s hard judge if the results are consistent across those architectures or the differences arise from the difference in datasets.
>
> A2: The results support the hypothesis by comparing the difference between the left end and the right end of a curve. We do not compare across models. Also, some models are vision models, and some are text models, so it is hard to use the same dataset.

---

### Author Response · Authors · 2022-11-19
**Thank you for all reviewers and we made updates.**

We thank all reviewers for the helpful comments.
We read them carefully multiple times and updated the paper with the following changes.

1. Added experiments with natural inputs (Appendix A and Figure 7,8,9). For vision experiments, we use images and predict foreground and background. For text experiments, we use amazon reviews and predict product categories and ratings. We also run ablation for different layer widths, applying dropout and mixup. The results still show that function sharing weakens systematic generalization. Also, it does not weaken training accuracy or i.i.d. generalization.

2. We extend the discussion of why function sharing happens (Appendix C). The neural network training process is complicated, so it is hard to describe what happens. Instead, we have a conjecture, which deepens the previous discussion.

3. Other changes. We made the variable names consistent in different sections. We point out that function sharing can also be called activation or feature sharing. We also made other detailed changes.

Due to the time limitation, we may be unable to address all the comments, but they will help us in our future work.

---

> ### Comment · Reviewer_hir1 · 2022-11-21
> **The paper has improved, but some key concerns remain.**
>
> Thanks to the authors for the updated paper and additional experiments.
>
> In my original review, I wrote that the datasets were adversarially generated, and raised two issues:
> 1) "they combine input stimuli in very unnatural ways (averaging images, or concatenating completely unrelated pieces of text)"
> 2) "and then enforce extremely strong correlations between these inputs at train time, which are completely reversed at test time."
>
> The new experiments with more realistic datasets address the first point, but not the second. So while these added experiments improve the paper, my overall concern about whether "this process has anything to do with any real-world data generating process" remains. In what situation will a model never see a single 5-star review of a book (for example), only of electronics, and then *only* see 5-star book reviews at test time? For an example of datasets which *actually* come from a real-world data generating process, and contain a distribution shift requiring generalization, without being adversarially sampled, the authors might consider e.g. WILDS (https://wilds.stanford.edu/).
>
> Furthermore, the results on the Appendix A datasets are not as compelling as the main results; for example, the vision networks show very little effect of increasing number of shared layers on test sample accuracy on the real datasets, while on the even-more adversarial ones reported in the main text the results are much stronger. Why not report the results on the realistic experiments in the main text, and put the other experiments in the supplement instead?
>
> Finally, I find that the rhetoric is still misleading, as I highlighted in my original review. The authors describe this as a "fundamental conflict" between deep learning and systematic generalization; however, I do not think they engage enough with the reasons that function sharing can be helpful. They only briefly mention that function sharing is related to auxiliary tasks, and do not really explore the extensive literature on how auxiliary tasks can support generalization. Engaging with this literature would, I think, require reflecting more seriously on the issues above about adversarially sampled datasets.
>
> Thus, while I think the paper has improved (particularly in incorporating the Appendix A experiments, but also including dropout, layer size etc.), I think it could be a much better contribution with a more nuanced perspective. I do think the experiments are interesting, and I hope the authors will take some of my suggestions whether this paper is accepted here or elsewhere. I will update my score accordingly.

---

> > ### Author Response · Authors · 2022-11-24
> > **Thank you for the comments and suggestions.**
> >
> > Here are some responses to the comments. We will take the suggestions and continue to improve the paper.
> >
> > The label combination design is based on the settings of systematic generalization, where a new output combines seen parts.
> > It might be less directly related to actual datasets or data generating processes.
> > However, systematic generalization is a critical ability in human learning.
> > For example, humans can infer 5-star book reviews from 5-star reviews of other categories and other book reviews. So this paper discusses why deep learning does not do something that humans can, and it might be helpful on the way to artificial general intelligence. Systematic generalization can also benefit fast and effective learning with less data. It might also be related to generating new objects by human imagination.
> >
> > The function sharing reduces the systematic generalization accuracies by half or more in the experiments. We would rearrange the paper and make it more clear.
> >
> > Function sharing is helpful and widely used in deep learning, and it is critical for the power of deep learning. However, the previous work mainly discusses i.i.d. generation, while we discuss systematic generalization. So this work indicates that function sharing is helpful for i.i.d. generalization but may not be for systematic generalization. We would rewrite the paper, include more advantages of function sharing, and clarify that we only discuss systematic generalization cases.

---

> > > ### Comment · Reviewer_hir1 · 2022-11-24
> > > **Comparison to human capabilities should be based on data, not researcher intuitions.**
> > >
> > > It may indeed be true that humans can generalize to 5-star book reviews from 5-star reviews of other categories, but I am not aware of anybody actually testing people's systematic generalization in cases in which the entirety of their learning experience exhibits very strong correlations, which are flipped at test time. Human learning experience is vastly richer and more varied than the datasets considered in this paper; yet humans also exhibit frequent (even systematic) errors. One of the key lessons of cognitive psychology is that our intuitions about when and how our minds work are often wrong. Thus, if AI researchers want to make a claim about human capabilities, it should be based on actual experiments in which actual humans exhibit the generalization in question (in domains they have never experienced before), rather than researcher intuitions about what humans can do.

---

> > > > ### Author Response · Authors · 2022-12-02
> > > > **Response to the comments**
> > > >
> > > > Thank you for the comments.
> > > > Here, we mean systematic generalization is a type of "ideal or desired ability." Also, from another perspective, classic symbol processing algorithms are supposed to be good at the type of generalization (Fodor and Pylyshyn 1988), and we discuss whether deep learning can do it.
> > > >
> > > > A strong correlation has also been used in different studies of systematic generalization.
> > > > For example, CFQ designs a different compound distribution: "the distribution of compounds in the train set is as different as possible from the distribution in the test set."
> > > > Also, in SCAN, a single word appears only in one context in training but in other contexts in the test.
> > > > The strong correlation might be more acceptable if we regard the test data contain both i.i.d. and o.o.d. data, but we focus on the o.o.d. data.
> > > > We will reconsider more details and make it more clear.

---

### Decision · Program_Chairs · 2023-01-20

**Decision:**

Reject

**Justification For Why Not Higher Score:**

Please see meta review

**Justification For Why Not Lower Score:**

The decision is reject.

**Metareview: Summary, Strengths And Weaknesses:**

This paper hypothesizes that systematic generalization is fundamentally at odds with the tendency of deep learning models to share sub-modules. It provides both a theoretical description and derivation of the issue as well as supporting experiments with several deep learning architectures.

Strength:
- The hypothesis introduced in this paper is new.
- The paper provides empirical evidence of why parameter sharing (function sharing) leads to performance drop in systematic generalization for deep learning models.


Weakness:
- The paper is not very well written;
- The problem formulation and experiment details are not satisfactory
- The initial experiments of the paper were not natural. During the rebuttal the authors added new experiments that should replace the bulk of the experiment section in the initial version. Also the following concern is not addressed in the new experiments that the adversarial generation process" enforces extremely strong correlations between these inputs at train time, which are completely reversed at test time."
- if AI researchers want to make a claim about human capabilities, it should be based on actual experiments in which actual humans exhibit the generalization in question
- In the ablation studies the authors use different datasets across those architectures thus it’s hard judge if the results are consistent across those architectures or the differences arise from the difference in datasets.

 The paper requires major re-write.
I invite the authors to consider the reviewers comments and discussions for improvement of the paper.